

# Comprehensive evaluation of methods for identifying tissues or cell types of origin of the plasma cell-free transcriptome

Tingyu Yang[1,2], Yulong Qin[1,2], Shuo Yan[1,2], Sijia Guo[2,3], Jinghua Sun[2], Jiayi Huang[2,4], Jiayi Li[1,2], Qing Zhou[2], Xin Jin[1,2,5,6] and Wen-Jing Wang[2]

[1] College of Life Sciences, University of Chinese Academy of Sciences, Beijing, China
[2] BGI Research, Shenzhen, China
[3] College of Life Sciences, Northwest University, Xi'an, China
[4] College of Life Sciences and Oceanography, Shenzhen University, Shenzhen, China
[5] Shenzhen Key Laboratory of Transomics Biotechnologies, BGI Research, Shenzhen, China
[6] The Innovation Centre of Ministry of Education for Development and Diseases, School of Medicine, South China University of Technology, Guangzhou, China

Corresponding authors
Xin Jin, jinxin@genomics.cn
Wen-Jing Wang,
wangwenjing@genomics.cn

## ABSTRACT

Plasma cell-free RNA (cfRNA) is derived from cells in various tissues and organs throughout the body and reflects the physiological and pathological conditions. Identifying the origins of cfRNA is essential for comprehending its variations. Only a few tools are designed for cfRNA deconvolution, and most studies have relied on traditional bulk RNA methods. In this study, we employed human tissue and cell transcriptomic data as reference sets and evaluated the performance of seven deconvolution methods on cfRNA. We compared the analysis results of cell types and tissues of origin of plasma cfRNA and chose to use single-cell RNA sequencing (scRNA-seq) data as reference to conduct further evaluation of deconvolution methods. Subsequently, we assessed the accuracy and robustness of the methods by utilizing simulated cfRNA data generated from scRNA-seq. We also evaluated the methods' accuracy on real plasma cfRNA data by analyzing the correlation between the predicted cell proportions and the corresponding clinical indicators. Moreover, we compared the methods' effectiveness in revealing the impacts of diseases on cells and evaluated the performance of cancer classification models based on the cell origin data they provided. In summary, our study provides valuable insights into cfRNA origin analysis, enhancing its potential in biomedical research.

# INTRODUCTION

Plasma cell-free RNA (cfRNA), released from cells through active secretion, apoptosis, or necrosis, reflects the health status of tissues and organs (*Koh et al., 2014*). Primarily sourced from blood cells, plasma cfRNA also includes genes specific to various other cell types and tissues (*Moufarrej et al., 2023*; *Ibarra et al., 2020*). Changes in specific tissue or cell activity, function, and quantity under disease conditions can cause abnormal fluctuations in the corresponding origin signals in plasma cfRNA (*Pös et al., 2018*). Research through tissues

of origin of plasma cfRNA has discovered increased lung and breast tissue signals in the plasma cfRNA of lung and breast cancer patients (*Larson et al., 2021*). Additionally, studies analyzing the cell types of origin of plasma cfRNA have identified increased liver cell signals in the plasma cfRNA of liver cancer patients (*Safrastyan, zu Siederdissen & Wollny, 2023*). Beyond reflecting changes in the tissue regions or cell signals where cancer is located, analyzing the cell types of origin of plasma cfRNA can also characterize changes in blood cells involved in immune regulation. Research has indicated that the signal from platelets in the plasma of lung cancer patients is increased (*Beck et al., 2019*). There are also studies that have built disease classification models based on the results of analyzing the cell types of origin of plasma cfRNA, which can effectively distinguish liver cancer patients from healthy individuals (*Safrastyan, zu Siederdissen & Wollny, 2023*). Analyzing the cell types or tissues of origin in plasma cfRNA can characterize the magnitude of cell or tissue signals in the plasma, infer the impact of diseases on cells, and classify diseases.

Many studies treat plasma cfRNA, which is a mixture composed of signals from multiple cell types, as bulk data for origin analysis. Some research employs gene set scoring methods to quantify specific tissue and cell signals in plasma cfRNA, such as using liver-specific genes for liver disease analysis and trophoblast-specific genes for preeclampsia studies (*Vorperian et al., 2022*; *Sun et al., 2023*). However, this strategy cannot provide a whole picture of the origins of cfRNA. Some studies use bulk deconvolution methods to generate gene expression profiles (GEPs) based on tissue or cell reference data and calculate the relative proportions of different tissue or cell origin signals in plasma cfRNA. Deconvolution methods such as CIBERSORTx (CSx) and non-negative matrix factorization (NMF) are employed to analyze cell types and tissues proportions in plasma cfRNA, utilizing single-cell data from Tabula Sapiens project (TSP) and tissue data from Genotype-Tissue Expression (GTEx), respectively (*Ibarra et al., 2020*; *Vorperian et al., 2022*; *Lonsdale et al., 2013*; *The Tabula Sapiens Consortium, 2022*).

Traditional deconvolution methods designed for bulk data analysis, typically involving fewer and clearer cell types, may not be as effective for plasma cfRNA, which originates from a diverse range of cell types and is prone to significant RNA degradation, leading to high gene absence rates. These factors may compromise the accuracy of such methods. A new method, Deconformer (*Yan et al., 2024*), has been developed by our research group specifically for body fluid cfRNA analysis. Utilizing pathway information, Deconformer trains on extensive simulated data based on deep learning, enhancing its ability to accurately identify the relative proportions of cell types, even despite high gene absence rates. Therefore, comparing the results of analyzing cell types and tissues of origin in plasma cfRNA, and systematically evaluating the performance of different types of deconvolution methods when applied to plasma cfRNA will provide valuable insights into cfRNA origin analysis.

In this study, we evaluated the performance of seven methods across three categories: deconvolution based on GEPs (CSx (*Newman et al., 2019*), AutoGeneS (*Aliee & Theis, 2021*), GEDIT (*Nadel et al., 2021*), MuSiC (*Wang et al., 2019*), SCDC (*Dong et al., 2020*)), scoring based on feature gene sets (xCell (*Aran, Hu & Butte, 2017*)), and deconvolution based on deep learning combined with pathway information (Deconformer (*Yan et al.,*

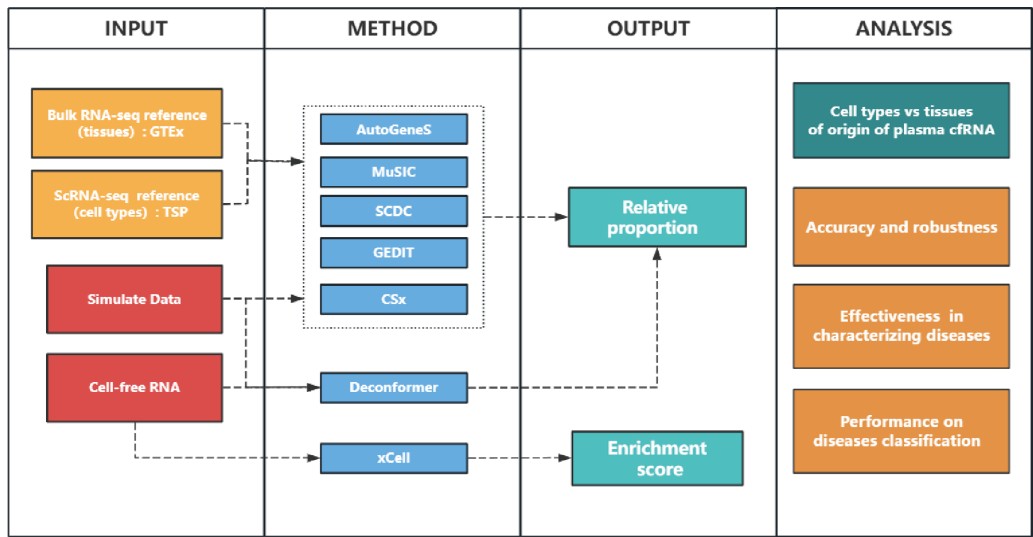

**Figure 1** **Study overview.** INPUT: We used bulk RNA-seq and scRNA-seq data as references to analyze cell-free RNA (cfRNA) and simulated data. GTEx: Genotype-Tissue Expression. TSP: Tabula Sapiens project. METHOD: We used CIBERSORT(CSx), AutoGeneS, GEDIT, MuSiC, and SCDC to analyze cfRNA based on input reference data, while Deconformer and xCell were based on built-in models or gene sets, respectively. OUTPUT: xCell generates enrichment scores, while other methods generate the proportions which sum to 1. ANALYSIS: We compared the major signal sources in the results of cell and tissue types of origin. We generated simulated data at different levels of gene detection to assess the accuracy and robustness of different methods in results of cell types of origin. We evaluated the accuracy of different methods when applied to plasma cfRNA, by comparing the correlation between the results of cell types of origin and clinical indicators. We also identified changes in signals of cell types in patients to evaluate their effectiveness in characterizing several diseases. Finally, we evaluated the performance of building cancer classification models by using the results of cell types of origin.

*2024*)). We compared the results of tissues and cell types of origin, evaluated the accuracy, robustness, and effectiveness of different methods in characterizing disease and classifying diseases. The flowchart is shown in Fig. 1 and the methods assessed are introduced in Table 1.

## MATERIALS & METHODS

### Deconvolution methods

We evaluated seven methods: CSx, AutoGeneS, GEDIT, MuSiC, SCDC, xCell, and Deconformer in our study. For AutoGeneS, GEDIT, MuSiC, SCDC, Deconformer, and xCell, analyses were conducted after installing the required Python modules or R packages locally. For CSx, the feature matrix was first generated through the online platform, and the deconvolution function obtained from their website was applied locally for deconvolution. All tools were run using default parameters.

xCell only provides scoring for specific cell types, rather than relative proportions. Moreover, xCell does not allow for reference file adjustments or scoring across different tissue origin ranges. As a result, this tool was not used when comparing cell types of origin with tissue of origin results, or in assessments using simulated data. Additionally, due to the

**Table 1  Overview of methods assessed.**

| Method | Approach | Custom reference | Reference input format | Gene Pre-selection | Algorithm | Output | Programming Language |
|---|---|---|---|---|---|---|---|
| AutoGeneS (*Aliee & Theis, 2021*) | GEP-based | Yes | h5ad | MOO | LR, Nu-SVR, NNLS | Predicted fraction | Python |
| GEDIT (*Nadel et al., 2021*) | GEP-based | Yes | dataframe | Filter by signature score | LR | Predicted fraction | Python |
| MuSiC (*Wang et al., 2019*) | GEP-based | Yes | rds | NA | W-NNLS | Predicted fraction | R |
| SCDC (*Dong et al., 2020*) | GEP-based | Yes | rds | NA | W-NNLS | Predicted fraction | R |
| CSx (*Newman et al., 2019*) | GEP-based | Yes | dataframe | Minimizing an inherent matrix property | $\nu$-SVR | Predicted fraction | R |
| Deconformer (*Yan et al., 2024*) | GEP&pathway –based | Yes | h5ad | Pathway-related | Transformer | Predicted fraction | Python |
| xCell (*Aran, Hu & Butte, 2017*) | Gene signature-based | No | NA | Marker gene | ssGSEA | Scores | R |

**Notes.**
MOO, multi-objective optimization; LR, Logistic Regression; NNLS, Non-Negative Least Squares; SVR, Support Vector Regression; ssGSEA, single-sample Gene Set Enrichment Analysis.

lack of specific cells around the lesion of colorectal cancer (CRC) and multiple myeloma (MM) in its built-in dataset, xCell was excluded from evaluations of how well deconvolution tools characterize cell impacts in these cancers. It was only used to compare the signal of hepatocytes and platelets and to build classification models based on deconvolution results. Deconformer only provides a cell-type deconvolution model based on the TSP dataset.

## Reference data preprocessing

At the tissue level, we obtained bulk RNA-seq data from the GTEx. Quality control was performed on the collected samples for each tissue by assessing gene detection counts and within-tissue sample correlations. We calculated the correlation of each sample with all other samples within the same tissue using Spearman's method, and then determined the correlation threshold for each tissue by subtracting five times the variance of these correlations from their mean value. Following the same criteria, we calculated the gene count threshold for each tissue. Samples with a mean correlation or gene count below these thresholds were considered as outliers and excluded. Tissue types with similar average gene expression, biological functions, and spatial proximity were merged. For example, exposed and non-exposed skin tissues were combined into a single skin category. Details of quality control and merging results are provided in supplementary tables (Table S1). Ultimately, we used bulk RNA-seq data of 31 tissues as reference.

At the cell type level, we utilized scRNA-seq data from the TSP. Following the approach by *Vorperian et al. (2022)* and *Yan et al. (2024)*, we combined cell types from TSP. We only kept cell types that had more than 10 cells, resulting in a total of 60 merged cell types. Due to the lengthy names of the merged cell types, we assigned each a representative name and included detailed correspondences in the supplementary tables (Table S2).
In order to reduce the time of analysis of CSx, AutoGeneS, GEDIT, MuSiC, and SCDC, we performed downsampling on the scRNA-seq and bulk RNA-seq data, sampling 100 samples per tissue or cell type, and using all available samples if fewer than 100 were present. The downsampled expression data were then converted into the appropriate file formats to be used as reference, according to the requirements of the various tools.

### Simulated cfRNA data

Simulated cfRNA data were generated from TSP as previously described (*Yan et al., 2024*) and were briefly summarized below. Cell types were randomly sampled from a pool of original types with equal probability. For each selected cell type, between 200 to 800 cells were randomly sampled for the mixture. If the total number of cells for a specific type was fewer than 200, all available cells were included. The mixture fractions for the cell types were specified by assigning a random ratio to each cell type, ensuring the sum of all ratios equaled 1. The expression data for each cell type was accumulated by first averaging the expression values for that cell type, then aggregating them into a weighted sum based on the mixture fractions. We generated 1,000 simulated cfRNA samples. To simulate data with different levels of gene detection, we randomly set 10%, 20%, 30%, 40%, and 50% of gene expression values to zero.

### cfRNA data preprocessing

The plasma cfRNA data were obtained from *Chen et al. (2022)*, *Roskams-Hieter et al. (2022)*, and *Tao et al. (2023)*. The plasma cfRNA data of patients with hepatitis B virus (HBV) and corresponding control was obtained from *Sun et al. (2023)*. These counts data were processed into TPM format, with sample quality control conducted according to the criteria outlined in the original articles. The *Chen et al. (2022)* dataset contains a total of 35 healthy samples, 21 esophageal carcinoma (ESCA) samples, 21 hepatocellular carcinoma (HCC) samples, 34 lung adenocarcinoma (LUAD) samples, and 36 stomach adenocarcinoma (STAD) samples. The *Roskams-Hieter et al. (2022)* dataset includes 30 healthy samples, 28 HCC samples, and 19 MM samples. The *Tao et al. (2023)* dataset comprises 50 healthy samples, 41 CRC samples, and 36 STAD samples. The *Sun et al. (2023)* dataset consists of 171 healthy samples and 40 HBV samples. To avoid the influence of batch effects from different datasets, when comparing the results of cell types of origin between disease and healthy individuals, we used the healthy individuals from the same datasets as the control group.

### Construction and evaluation of classification models

We used the results of cell types of origin as input features to build classification models. Each dataset was split into a training set and validation set at 7:3 ratio. We generated 100 different training-validation set combinations through random sampling. Given the class imbalance present in some datasets, we applied synthetic minority over-sampling technique (SMOTE) to upsample both the training and validation sets. A random forest model was used for training and prediction, and the area under the curve (AUC) was recorded for each model's prediction on the validation set. The classification performance of different tools based on their results of cell types of origin was evaluated by averaging the

AUCs from the 100 models. To compare the effectiveness of cancer classification models constructed using the results of cell types of origin from different tools, we defined a classification performance score. This score was calculated for each cancer type within each dataset by subtracting the average AUC of all tools from the average AUC of a given tool, then dividing the result by the variance of the AUCs from all tools.

## Statistical analysis

When comparing the accuracy of different tools applied to simulated cfRNA, we used root mean square error (RMSE) and concordance correlation coefficient (CCC), calculated in the same way as in previous studies using R (*Menden et al., 2020*). For evaluating the correlation between deconvolution results and clinical indicators, we applied Spearman's correlation, setting a significance threshold of $p$-value $< 0.01$ and absolute correlation coefficient $|r| \geq 0.2$. In analyzing differentially expressed genes from datasets of various sources and comparing the results of cell types of origin across groups, statistical differences were assessed using the Wilcoxon rank-sum (Mann–Whitney U) test. We calculated the Rank Gap based on the results of the rank-sum test, which quantifies the degree of difference in platelet signals between groups.

## RESULTS

### Comparing the results of cell types of origin and tissues of origin in plasma cfRNA

We obtained expression profiles of different types of cells and tissues from the TSP (*The Tabula Sapiens Consortium, 2022*) and the GTEx (*Lonsdale et al., 2013*) as reference datasets to conduct both tissues of origin and cell types of origin analysis on plasma cfRNA from healthy individuals across four datasets.

For the results of cell types of origin, we found that the overall signal proportion of blood cells ranges from 67% to 93%, making them the primary sources of plasma cfRNA (Fig. 2A). Upon further examination of specific cell signal proportions, we discovered that platelet signals are the highest (exceeding 31%), followed by red blood cells, NK cells, monocytes, and other blood-derived cells (Fig. 2B). As for the results of tissues of origin, we observed that the proportion of whole blood is less than 20%, which is significantly lower than previous research findings (*Koh et al., 2014*; *Ibarra et al., 2020*), while the proportion of spleen is high in most of the methods. We conducted the same analysis on plasma cfRNA from healthy individuals in other datasets, and the results were similar (Fig. S1). As we know, whole blood and spleen share a group of cell types, the contribution of whole blood could be calculated on the spleen.

In summary, by comparing the results of cell types of origin and results of tissues of origin in plasma cfRNA from healthy individuals, we believe that the results of cell types of origin are more rational when using deconvolution methods based on single-cell data to analyze cfRNA. Therefore, all subsequent evaluations will be based on the strategy of analyzing the cell types of origin.

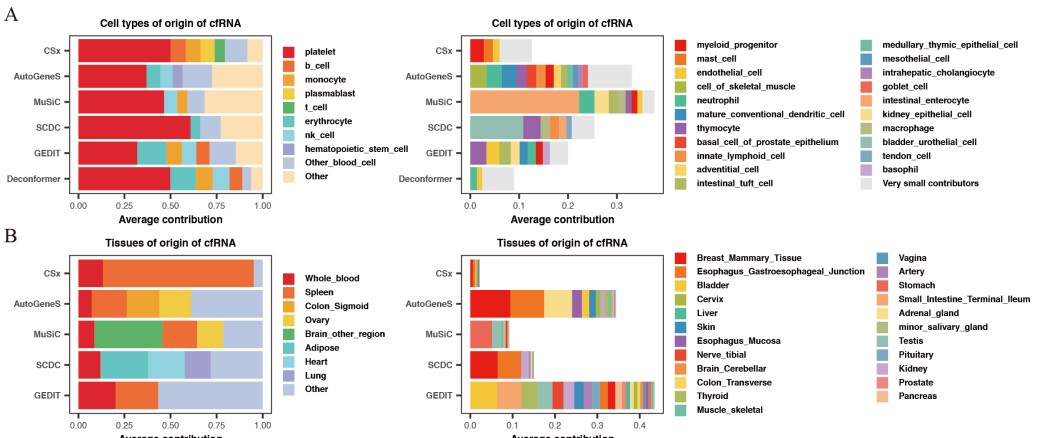

**Figure 2** Comparing the results of cell types of origin and tissues of origin in plasma cfRNA. (A) Cell types of origin, displaying proportions of each major blood-derived cell type (> 5%) on the left, while other blood cell types (<5%) and other cell types (non-blood) are shown on the right. (B) Tissues of origin, displaying proportions of whole blood and major tissue types on the left, while other tissue contributions on the right. X-axis: origin proportions, Y-axis: different methods, different colors represent different cell or tissue types.

## Assessment of accuracy and robustness across methods

To evaluate the accuracy of different methods for analyzing the cell types of origin, we generated simulated data based on the TSP dataset by mixing expression profiles of various cell types in different proportions. Additionally, we noted that the initially generated simulated data had an average gene detection count of 17,000, while the gene detection rates in real plasma cfRNA data are much lower (approximately 8,000–14,000). To make the simulated data more closely resemble the detection conditions of actual plasma cfRNA, we set different gene missing rates in the simulated data. The simulated data with a missing rate of 30–50% closely matches the gene detection conditions of actual plasma cfRNA (Fig. S2).

We conducted cell type origin analysis on the simulated data and compared the predicted proportions to the actual proportions to calculate the CCC (Fig. 3A) and RMSE (Fig. 3B) for the results of different methods. These metrics indicate the accuracy of the cell types of origin analysis by different methods. We found that as the gene missing rate increases, the CCC of most methods decreases, and the RMSE increases. As the missing rate increases from 0% to 50%, MuSiC (median CCC maintains 0.87, median RMSE maintains 0.02) and Deconformer (median CCC decreases from 0.99 to 0.96, median RMSE increases from 0.005 to 0.011) are minimally affected by gene dropout. In contrast, other methods are more significantly affected by the missing rate, such as AutoGeneS (CCC decreasing from 0.81 to 0.28, RMSE increasing from 0.02 to 0.045), CSx (CCC decreasing from 0.98 to 0.59, RMSE increasing from 0.007 to 0.037). We found that for simulated data with gene detection numbers more closely to actual plasma cfRNA data (dropout rate of 0.3), Deconformer performed the best (median CCC = 0.98, median RMSE = 0.006), followed

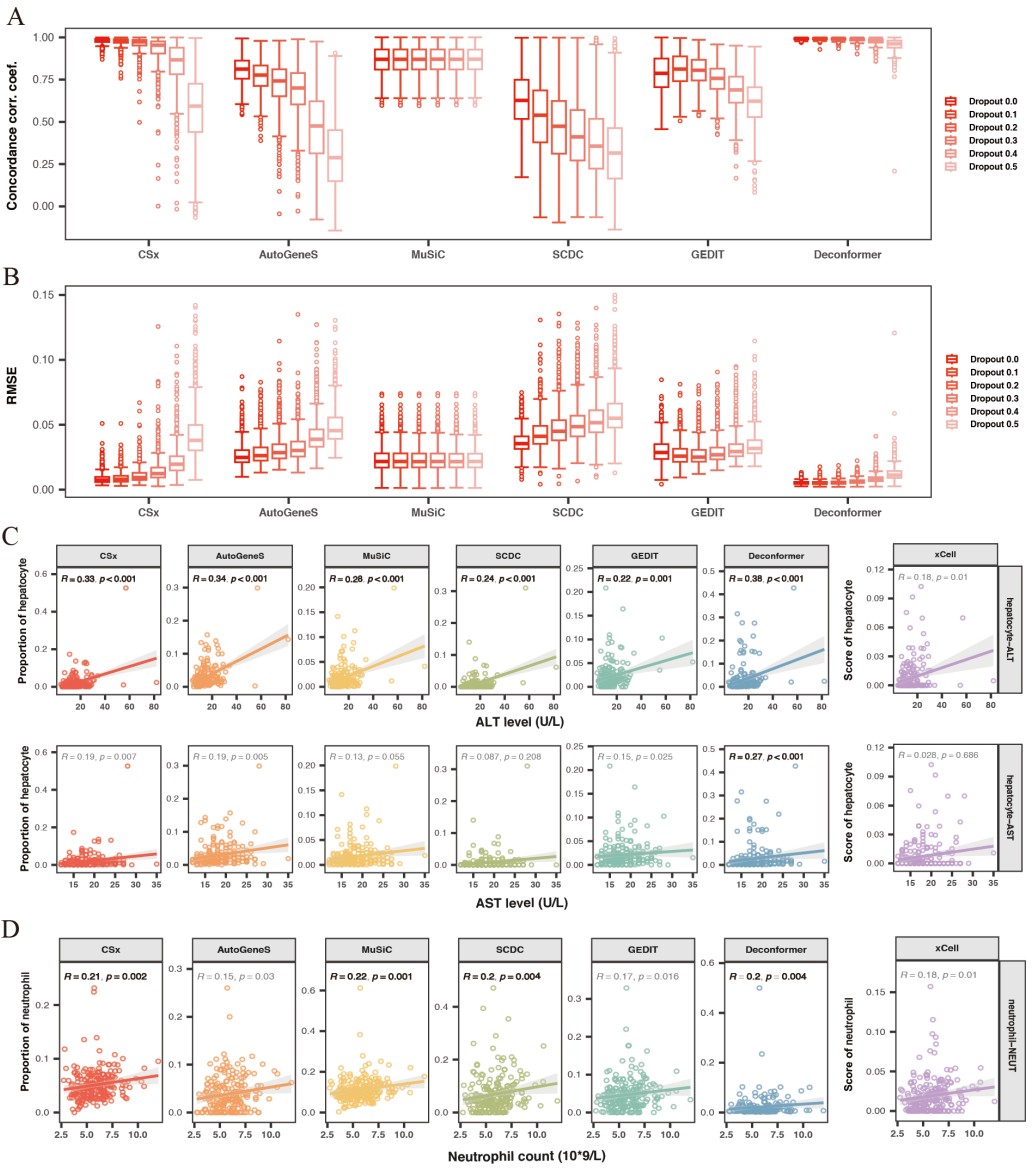

**Figure 3  Evaluating the accuracy and robustness of methods.** (A, B) Evaluating the accuracy and robustness of methods on simulated data. (A) Concordance correlation coefficient (CCC) and (B) Root mean square error (RMSE) between predicted proportions and actual proportions for different methods analyzing data with varying dropout rates. The intensity of the colors represents the levels of dropout. (C) Correlation between hepatic signals and liver injury indicators. The $X$-axis shows levels of Alanine Aminotransferase (ALT) and Aspartate Aminotransferase (AST), while the $Y$-axis represents hepatocyte signal levels. (D) Correlation between neutrophil signals and neutrophil counts. The $X$-axis indicates neutrophil counts, and the $Y$-axis represents neutrophil signal levels. Results with significant correlations ($R \geq 0.2$, $p$-value < 0.01) are highlighted in bold.

by CSx (median CCC = 0.95, median RMSE = 0.01) and Music (median CCC = 0.87, median RMSE = 0.02).

Across different levels of gene missing rates, Deconformer consistently shows the best performance, especially on data with lower gene detection numbers, maintaining high accuracy. While CSx performs well at lower missing rates, its accuracy is significantly affected as the missing rate increases, with a noticeable decline in performance when the missing rate reaches 50%. Therefore, we believe CSx is not suitable for data with low gene detection numbers. Compared to these two methods, MuSiC has shown the highest robustness, likely due to its approach of assessing weights across all genes without pre-selection. Our results indicate that Deconformer, CSx, and MuSiC demonstrate higher accuracy when applied to data similar to plasma cfRNA.

## Evaluating the correlation between the results of cell types of origin and clinical indicators

Previous studies have indicated that elevated levels of hepatocyte injury indicators suggest increased hepatocyte damage, which leads to the release of more cfRNA, thereby increasing the hepatocyte signal in the blood (*Sun et al., 2023*). To evaluate the accuracy of different methods in analyzing the cell types of origin in real plasma cfRNA data, we calculated the correlation between the hepatocyte signals in the origin results and liver injury indicators Alanine Aminotransferase (ALT) and Aspartate Aminotransferase (AST) (Fig. 3C). We found that hepatocyte signals from all methods showed a positive correlation with liver injury indicators. Among these, only Deconformer's hepatocyte signals were significantly correlated with levels of AST ($R \geq 0.2$, $p$-value $< 0.01$), and showed the highest correlation with ALT. Other methods that performed well in correlation with ALT include AutoGeneS (0.34), CSx (0.33), and MuSiC (0.29).

Blood cells are the primary source of plasma cfRNA. We hypothesized that an increase in the number of blood cells might lead to a rise in corresponding cell signals in the plasma. We also assessed the correlation between neutrophils (Fig. 3D), monocytes, basophils, erythrocytes, platelets, and their corresponding counts. We found that in the results of all methods, the signal from neutrophils positively correlated with neutrophil counts. These methods that showed significant positive correlations are: MuSiC (0.22), CSx (0.21), Deconformer (0.2), and SCDC (0.2). On the other hand, we found that all methods did not show significant correlations between signals of other cell types and their corresponding counts (Fig. S2).

Overall, Deconformer, MuSiC, and CSx demonstrated high consistency in analyzing the cell types of origin in real plasma cfRNA data with clinical indicators.

## Evaluating the effectiveness of methods in characterizing the impact of liver diseases

To evaluate the effectiveness of these methods in characterizing the impact of liver disease on cells, we collected datasets related to liver diseases. We analyzed plasma cfRNA data from patients with HBV infection or HCC (HBV data from *Sun et al. (2023)*, HCC data from *Chen et al. (2022)* and *Roskams-Hieter et al. (2022)*) using seven methods to analyze the cell types of origin. We examined the changes in signals of various cell types in the plasma cfRNA of liver diseases patients as indicated by the results of different methods.

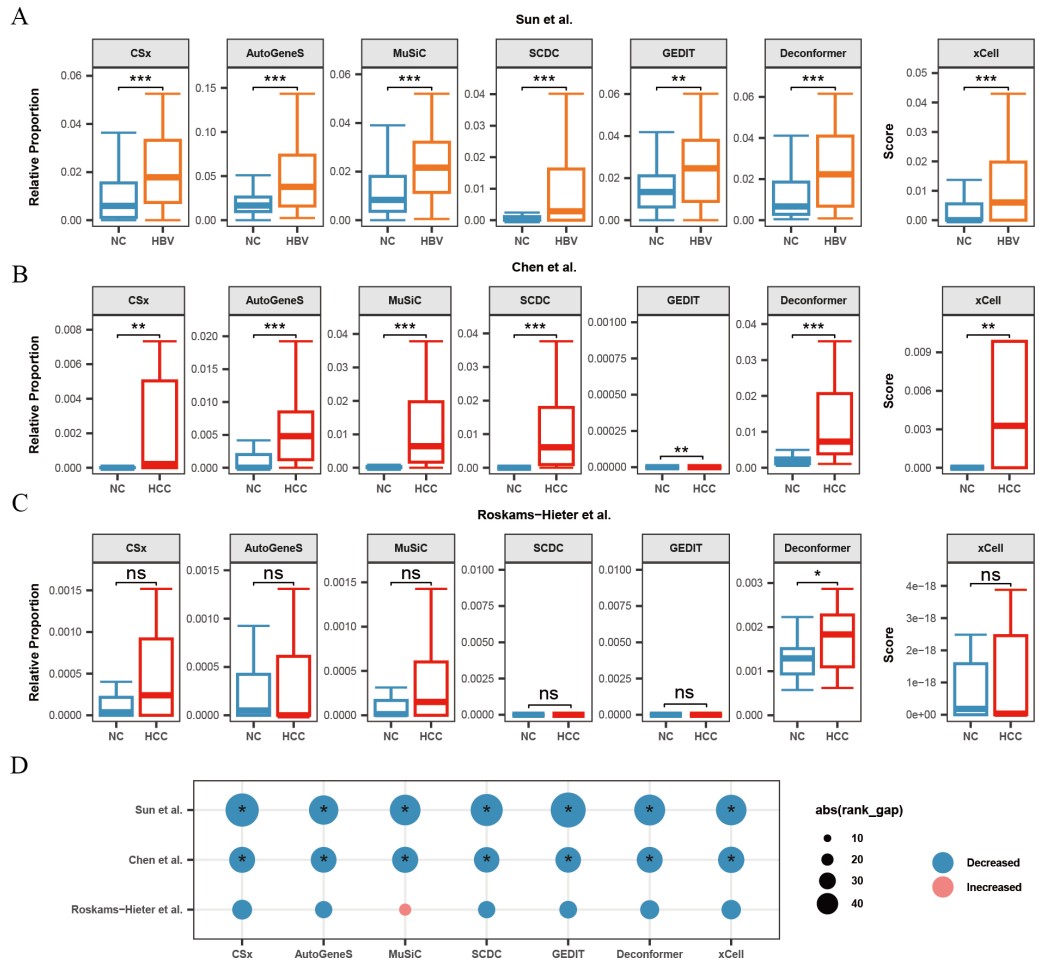

**Figure 4** **Changes in signals of hepatocytes and platelets in plasma cfRNA from liver disease patients.**
(A) Changes in the relative proportions and scores of hepatocytes in plasma cfRNA in hepatitis B virus
(HBV) patients from the *Sun et al. (2023)* study. (B, C) Changes in the relative proportions and scores
of hepatocytes in plasma cfRNA in hepatocellular carcinoma (HCC) patients from (B) *Chen et al. (2022)*
study and (C) *Roskams-Hieter et al. (2022)*. (D) Changes in proportions and scores of platelet in plasma
cfRNA in liver disease patients. The absolute value of the rank gap (abs(rank_gap)) represents the magni-
tude of signal change. The significance of differences is assessed using the Wilcoxon rank-sum test, with *p*-
values denoted as: *<0.05; **<0.01; ***<0.001. ns: not significant. NC, Normal control.

For both HBV and HCC (*Chen et al., 2022*), results from all methods demonstrated a
significant increase in hepatocyte signals in the plasma cfRNA of patients (Fig. 4A, 4B).
Previous research has shown that hepatocytes in patients with HBV and HCC are often
in a damaged state, and increased hepatocyte apoptosis could lead to the release of more
cfRNA into the plasma, thus elevating hepatocyte signals in plasma cfRNA. However,
in the *Roskams-Hieter et al. (2022)* dataset, while most methods showed an increase in
hepatocyte signals in the plasma cfRNA of HCC patients, only the cell type origin results
from Deconformer exhibited a significant increase in hepatocyte signals (Fig. 4C).

Additionally, we observed in all three liver disease datasets that the results from nearly all methods indicated a decrease in platelet signals in the plasma cfRNA of liver disease patients (Fig. 4D). Platelets are a major source of signals in plasma cfRNA and are involved in immune regulation. In the *Roskams-Hieter et al. (2022)* dataset, the decrease in platelet signals was noted but not significant.

We found that, compared to the other two datasets, the changes in hepatocyte and platelet signals in the plasma of patients from *Roskams-Hieter et al. (2022)* were lower, regardless of the method used. Further analysis of differentially expressed genes in the plasma cfRNA profiles revealed that the *Roskams-Hieter et al. (2022)* dataset has fewer differentially expressed genes (DEGs) in HCC patients, particularly lacking up-regulated liver-specific and down-regulated platelet-specific DEGs (Table S3). We speculate that the low degree of gene expression differences might result in less significant changes in cell signals in the results of methods based on GEPs or feature gene sets. On the other hand, Deconformer, which incorporates pathway information in its analysis, shows higher sensitivity.

## Evaluating the effectiveness of methods in characterizing the impact of other cancers

To evaluate the performance of different methods for analyzing the cell types of origin in other cancer datasets, we further analyzed plasma cfRNA data from patients with various types of cancer, including CRC, STAD, MM, ESCA, and LUAD. Using six methods to analyze the cell types of origin, we examined the changes in signals from various cell types in the plasma cfRNA of liver disease patients across different methods.

When analyzing the results of cell types of origin in plasma cfRNA from CRC patients and healthy individuals, we found that AutoGeneS, CSx, and Deconformer robustly captured signals from intestinal enterocytes or immature enterocytes across different datasets (*Chen et al., 2022*; *Tao et al., 2023*), with consistent trends in cell signal changes across these datasets (Fig. 5A, 5B). Specifically, the cell types of origin results from CSx and AutoGeneS showed a significant decrease in intestinal enterocyte signals in the plasma cfRNA of CRC patients, while Deconformer's results showed a significant decrease in immature enterocyte signals in the plasma of CRC patients. In the results from most other methods, the signals from these two cell types were extremely low. Existing research indicates that the normal differentiation of intestinal stem cells in CRC patients is hindered, leading to a reduction in the number of immature intestinal cells and intestinal cells derived from these stem cells, which we believe could be the cause of the reduced signals in the plasma (*Liang et al., 2022*).

When analyzing the results of cell types of origin in plasma cfRNA from MM patients and healthy individuals, we observed that the results from all methods indicated a decrease in B-cell signals and a significant increase in red blood cell signals (Fig. 5C). MM is a malignant tumor affecting the blood and bone marrow. In MM patients, abnormal B cells accumulate in the bone marrow and form tumors. These abnormal B cells undergo significant changes in their expression profiles and fail to perform normal immune functions. The reduction in the proportion of normal B cells might explain the decreased B-cell signals in the

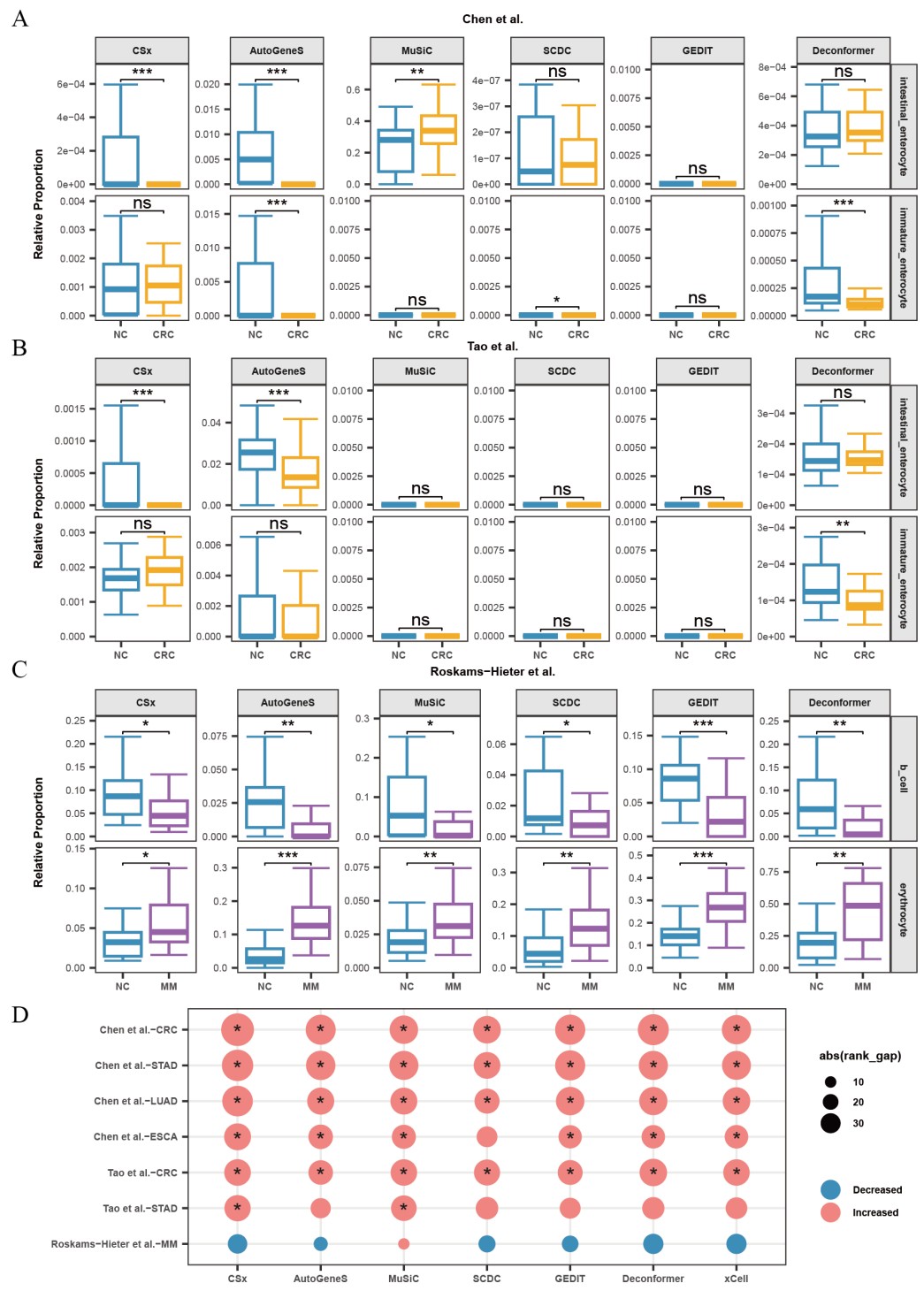

**Figure 5  Changes in proportions of cell types in plasma cfRNA from patients with several cancers.** (A, B) Changes in the relative proportions of intestinal-related cells in plasma cfRNA from colorectal cancer (CRC) patients from *Chen et al. (2022)* and (B) *Tao et al. (2023)*. (C) Changes in the relative proportions of B cells and erythrocytes in plasma cfRNA from multiple myeloma (MM) (continued on next page...)

**Figure 5 (…continued)**
patients from *Roskams-Hieter et al. (2022)*. (D) Changes in the relative proportions of platelets in plasma cfRNA from several cancer patients. The absolute value of the rank gap (abs(rank_gap)) represents the magnitude of signal change. The significance of differences is assessed using the Wilcoxon rank-sum test, with $p$-values denoted as: $*<0.05$; $**<0.01$; $***<0.001$. ns: not significant. NC, Normal control.

plasma (*Liu et al., 2021*). Additionally, the progression of MM increases erythroid cell apoptosis, releasing more cfRNA into the plasma, which could be the cause of the increased red blood cell signals in the plasma (*Moyo et al., 2015*).

For other types of cancer, the reference data does not include cell types from the tumor regions or cell types specifically affected by the cancer. Platelets are affected by almost every type of cancer. We characterized the changes in platelet signals in the plasma cfRNA from patients with various cancers including CRC, STAD, ESCA, LUAD, and MM (Fig. 5D). We found that for patients with CRC, STAD, ESCA, and LUAD, the results of cell types of origin of plasma cfRNA from all methods indicated an increase in platelet signals, with most of these increases being significant. Previous research has shown that cancer significantly affects platelets, with increased platelet production and activity observed in patients with gastric cancer, lung cancer, esophageal cancer, *etc.* These effects of cancer on platelets are likely the cause of changes in platelet signals in plasma cfRNA. We found that for patients with STAD from *Tao et al. (2023)*, only the results from MuSiC and CSx showed a significant increase in platelet proportions.

## Evaluating the effectiveness in cancer classification

To evaluate the effectiveness in cancer classification, we utilized the cell proportions or scores generated by methods as input features to classify between cancer patients and healthy individuals across various datasets. We assessed the classification performance of the models by calculating the average AUC from models generated through multiple random samplings.

Our results indicate that for most classification tasks, the results of cell types of origin from most methods demonstrate good classification performance. We observed that the best-performing methods vary across different classification tasks (Figs. 6A–6C). Specifically, for HCC data from *Chen et al. (2022)* (Fig. 6A), the results of cell types of origin from Deconformer (AUC: 0.81) and xCell (AUC: 0.80) performed best. For CRC data from *Tao et al. (2023)* (Fig. 6B), CSx (AUC: 0.87) and Deconformer (AUC: 0.85) performed best. For CRC data from *Chen et al. (2022)* (Fig. 6A), AutoGeneS (AUC: 0.87) and Deconformer (AUC: 0.87) performed best. The details of classification performance (AUC, sensitivity, specificity, standard deviation of AUC) are provided in the supplementary tables (Tables S4–S6).

We found that when using results of cell types of origin to construct models, the classification performance for the same cancer type varies significantly across different datasets (Fig. 6D). For example, in the HCC data from *Chen et al. (2022)*, the average AUC across different methods reached as high as 0.9, whereas in the HCC data from *Roskams-Hieter et al. (2022)*, the average AUC was less than 0.7. This lower classification performance reflects smaller differences in cell origin results between groups. Additionally, we observed

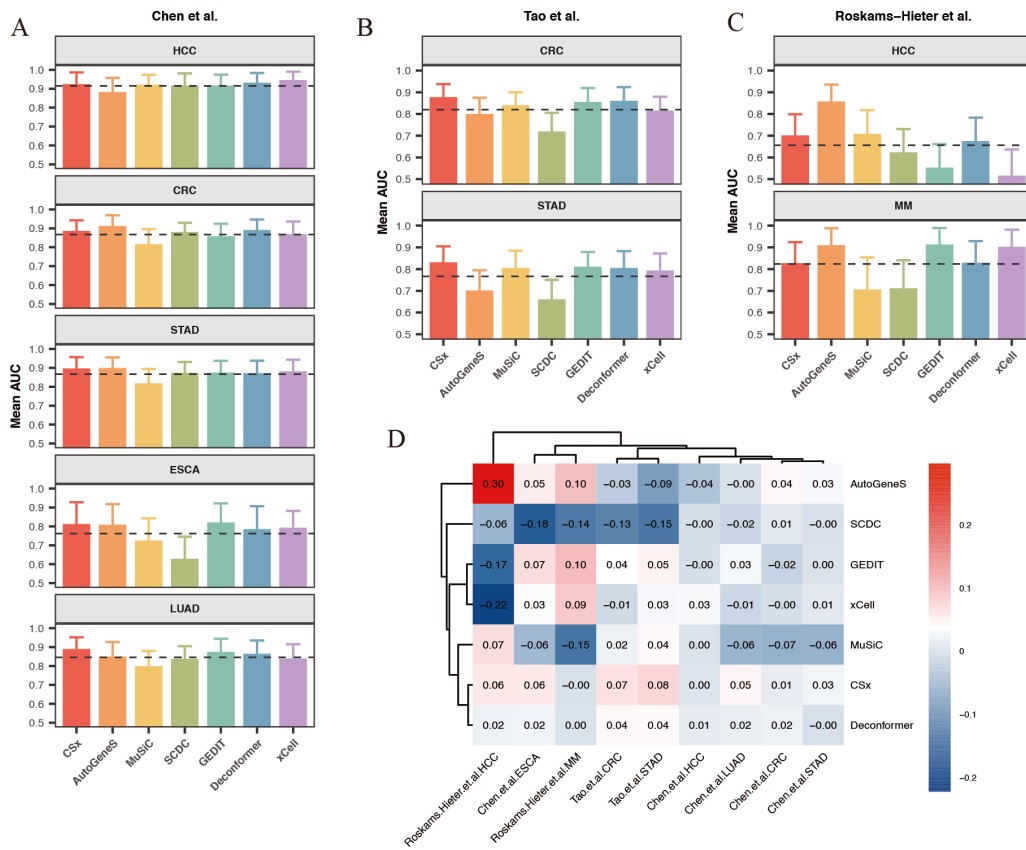

**Figure 6  Comparison of cancer classification model performance based on cell types of origin results.** (A–C) Average area under the curve (AUC) values for cancer classification models constructed for five types of cancer from (A) *Chen et al. (2022)* and (B) *Tao et al. (2023)* (C) *Roskams-Hieter et al. (2022)*. The error bars are calculated based on the mean AUC ±the standard deviation of the AUC. (D) Relative performance of models constructed using different methods for different types of cancer from various sources, with a color gradient from red to blue indicating relative performance from high to low.

that within the same dataset, the classification performance varies between different cancers. HCC generally showed the best classification results, while ESCA showed the worst. This variation may be due, in part, to the cell types of origin possibly not encompassing the main cell types affected by the cancer; on the other hand, different cells are affected differently by cancer, leading to varying degrees of change in cfRNA cell signals. Given that both cancer type and dataset can indirectly influence classification performance, we assessed the relative effectiveness of different methods for results of cell types of origin in each classification task. To do this, we normalized each method's AUC against the average AUC for all methods within each classification task, generating a classification effectiveness score. A score above zero indicates that the method's classification performance is above the average level, and vice versa. The best-performing method varied across different classification tasks. For CRC in the *Chen et al. (2022)* dataset, AutoGeneS showed the highest classification effectiveness score of 0.3. In the *Roskams-Hieter et al. (2022)*. dataset for MM, GEDIT showed the highest score of 0.1. When considering the overall performance of each method

across all classification tasks, we found that CSx and Deconformer consistently achieved classification effectiveness scores of zero or higher for all cancer types across all datasets. In contrast, other methods showed less robust performance across different classification tasks, sometimes scoring very low, such as SCDC with a score of −0.18 for ESCA in the *Chen et al. (2022)* dataset and xCell with a score of −0.22 for HCC in the *Roskams-Hieter et al. (2022)* dataset.

In summary, by evaluating the effectiveness of different methods' results applied to constructing cancer classification models, we found that using results of cell types of origin as features provides good classification performance for distinguishing between cancer patients and healthy individuals. CSx and Deconformer, in particular, showed consistently good classification performance across various datasets.

## DISCUSSION

Analyzing the origins of plasma cfRNA is a computationally challenging problem. Although past studies have made some attempts, they lack a comparative evaluation of the effectiveness of using different strategies and methods. Our research compared results of tissues of origin and results of cell types of origin, systematically assessing the accuracy and robustness of various types of methods applied to the cell types of origin of plasma cfRNA. Furthermore, our study evaluated how well these methods characterize the impact of diseases on cells and their performance in constructing cancer classification models.

Previous studies have already demonstrated that plasma cfRNA primarily originates from blood cells (*Ibarra et al., 2020*). We found in our results of cell types of origin that blood cells are the predominant source, accounting for over 70% of the total, whereas in the results of tissues of origin, the proportion of whole blood is much lower, less than 25%. The discrepancy between the two strategies might be due to the high similarity in expression profiles among different tissues, which often contain many of the same types of cells.

Many studies have noted that high correlation in reference datasets can diminish the effectiveness of deconvolution (*Avila Cobos et al., 2020*; *Sturm et al., 2019*; *Sutton et al., 2022*). Additionally, the methods we evaluated were originally developed for deconvolution based on single-cell data, and may not be ideally suited for using tissue bulk data as reference datasets, leading to poorer deconvolution results. Furthermore, the presence of various types of blood cells in blood means that cell types of origin can more precisely quantify the proportions of different types of blood cells, such as platelets, red blood cells, and various immune cells, offering a higher resolution analysis compared to tissues of origin. Therefore, we recommend employing a strategy of using results of cell types of origin.

In our evaluation of method accuracy and robustness using simulated data, we found that Deconformer developed by our research group exhibited the highest accuracy and was least affected by gene dropout rates. CSx also performed well overall, but its accuracy significantly decreased as gene dropout rates increased. Although MuSiC has the best robustness, being almost unaffected by gene dropout, its accuracy is lower than the

first two methods when analyzing simulated data with gene detection counts closest to typical cfRNA levels. When assessing the correlation between the results of cell types of origin and clinical indicators, we observed that Deconformer, CSx and MuSiC showed a relatively high correlation with clinical indicators. In evaluating the effectiveness of methods in characterizing the impact of disease on cells, we noted that most methods could identify an increase in hepatocyte signals in the plasma cfRNA of patients from *Sun et al. (2023)* HBV dataset and *Chen et al. (2022)* HCC dataset. However, for the *Roskams-Hieter et al. (2022)* HCC dataset, which had smaller differences in expression profiles, only Deconformer was able to detect an increase in hepatocyte signals. We also discovered that CSx, Autogene and Deconformer could characterize the decrease in signals of immature enterocytes or enterocytes in the plasma cfRNA of CRC patients from different datasets. When evaluating the effectiveness of constructing cancer classification models, we found that Deconformer and CSx consistently maintained good predictive performance across different datasets and cancer types, demonstrating their robustness and utility in cancer classification tasks.

During the evaluation process, in addition to the choice of methods, we identified other factors that could potentially affect the results of cell types of origin analysis of plasma cfRNA. Firstly, the gene detection count in plasma cfRNA influences the accuracy of cell types of origin results. Our findings suggest that a lower gene detection count in plasma cfRNA can lead to reduced accuracy of cell types of origin results. Deconformer and MuSiC have the highest robustness, which may mitigate the impact of this factor, resulting in better performance on data with lower gene detection count, compared to other methods.

Additionally, the degree of difference in the cfRNA expression profiles between diseased and healthy individuals affects the effectiveness of cell types of origin results in characterizing cell signal changes and constructing classification models. We found that, among HCC patients, compared to data from *Chen et al. (2022).*, the analysis of HCC patients' plasma cfRNA cell types of origin results from *Roskams-Hieter et al. (2022)* showed smaller changes in hepatocyte and platelet signals, and the classification models constructed based on these cell types of origin results also performed poorly. The dataset from *Roskams-Hieter et al. (2022)* exhibited fewer differences in cfRNA expression in HCC patients. Thus, we speculate that the low degree of gene expression differences can lead to poorer performance in characterizing disease-related cell signal changes and constructing disease classification models. Deconformer and CSx have the highest sensitivity, which may mitigate the impact of this factor, resulting in better performance on datasets in this situation, compared to other methods. However, the choice of reference datasets and the scope of cells analyzed according to the research goal, the normalization methods for plasma cfRNA expression profiles, and the various parameters set during the use of the methods can all affect the effectiveness of analyzing the origins of plasma cfRNA, which were not evaluated in this study.

## CONCLUSIONS

Overall, our study provides a systematic evaluation of strategies and methods for analyzing the origins of plasma cfRNA. Regarding strategies, we recommend adopting cell types

of origin analysis; for methods, we suggest using Deconformer or CSx for analysis. We identified factors that affect the effectiveness of cell types of origin analysis, including that lower gene detection counts can reduce method accuracy, and minimal differences in cfRNA expression profiles between diseased and healthy individuals can result in outcomes that fail to characterize the impact of the disease on cells. In summary, our research advances the understanding and application of plasma cfRNA.

## ACKNOWLEDGEMENTS

The authors thank members of the research group in BGI for helpful discussions.

### Funding
This research was funded by the Science, Technology and Innovation Commission of Shenzhen Municipality (JCYJ20170412152854656). The funders had no role in study design, data collection and analysis, decision to publish, or preparation of the manuscript.

### Grant Disclosures
The following grant information was disclosed by the authors:
The Science, Technology and Innovation Commission of Shenzhen Municipality: JCYJ20170412152854656.

### Competing Interests
Tingyu Yang, Yulong Qin, Shuo Yan, Sijia Guo, Jiayi Huang, Jiayi Li are students jointly trained by the universities and BGI Research. Jinghua Sun, Qing Zhou, Xin Jin and Wen-Jing Wang are employed by BGI Research.

### Author Contributions
- Tingyu Yang conceived and designed the experiments, analyzed the data, prepared figures and/or tables, authored or reviewed drafts of the article, and approved the final draft.
- Yulong Qin analyzed the data, prepared figures and/or tables, and approved the final draft.
- Shuo Yan analyzed the data, authored or reviewed drafts of the article, and approved the final draft.
- Sijia Guo analyzed the data, prepared figures and/or tables, authored or reviewed drafts of the article, and approved the final draft.
- Jinghua Sun conceived and designed the experiments, authored or reviewed drafts of the article, and approved the final draft.
- Jiayi Huang analyzed the data, prepared figures and/or tables, and approved the final draft.
- Jiayi Li analyzed the data, authored or reviewed drafts of the article, and approved the final draft.

- Qing Zhou conceived and designed the experiments, authored or reviewed drafts of the article, and approved the final draft.
- Xin Jin conceived and designed the experiments, authored or reviewed drafts of the article, and approved the final draft.
- Wen-Jing Wang conceived and designed the experiments, authored or reviewed drafts of the article, and approved the final draft.

## Data Availability

The raw data of results of tissues or cell types of origin of the plasma cell-free transcriptome and the script used to calculate AUC are available in the Supplemental File.

## Supplemental Information

Supplemental information for this article can be found online at http://dx.doi.org/10.7717/peerj.19241#supplemental-information.

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
