# Peer review of "Comprehensive evaluation of methods for identifying tissues or cell types of origin of the plasma cell-free transcriptome"

_PeerJ, doi:10.7717/peerj.19241_

## Round 0.1 · original submission · Minor Revisions

Please incorporate all reviewer comments into the revised manuscript and provide a detailed, point-by-point response in a separate document.

Reviewer 1 ·

Basic reporting

Cell-free RNA (cfRNA) analysis is important part of liquid biopsy studies. Methods for tissue/cell-type deconvolution would be of strong interests of the field. This study compared 7 previously published deconvolution methods and summarized the results. The manuscript and supplementary data provided by the authors were appropriate and informative to reproduce the analysis.

Experimental design

Comparisons of 7 analysis tools were performed in multiple approaches. Following two points are suggested although these are not essential.
- Figure 2 analysis provided general overview of how different tools works for tissue/cell-type deconvolution of cfRNA data. However, Deconformer and xCell were not compared in this analysis due to different requirements of reference cell/tissue RNA expression data. The authors could provide additional panels to show how Deconformer and xCell work if the same cfRNA data is analyzed.
- In Figure 2, “other” appeared to include many minor cell/tissue types contributing to cfRNA profile. For certain studies, detection of minor cell/tissue types are important. The authors could provide separate graph or pie chart (or other types of visualization) to show the composition of “other”.

Validity of the findings

The authors used standard parameters for tool usage. The results should be valid for these analysis purposes, for the cases in which tissue contribution is relatively large (>10% or more), for mainly cancer-derived cfRNA contribution.

Reviewer 2 ·

Basic reporting

1) The manuscript is clear and professionally written but contains minor typos and missing words that should be addressed. For example, there is a missing word in line 49, a missing verb in line 146, and a typo in line 147 (“corresponding”).

2) The paper provides sufficient background and appropriately cites relevant literature.

3) The article is well-structured and adheres to publication formats. Figures and tables are relevant, but some legends lack sufficient detail (see below). Additionally, the resolution of the main figures is quite low and should be improved. The resolution of supplemental figures is adequate, so this may be related to a conversion step during manuscript preparation.
- Legends for supplemental tables: Some legends appear to be missing. For instance, Supplemental Table 5 in particular would benefit from a legend, e.g. to clarify term “cor_the”—what type of correlation is used, and is it the mean of pairwise sample correlations?
- Supplemental Figure S2: Please specify the correlation method used. Additionally, some figure panels in S2 contain a small rectangle with a cross that appears to be an error and should be removed.

4) The submission is self-contained.

Experimental design

1) The research aligns well with the journal's aims and scope, addressing an important knowledge gap regarding the best approach for deconvolving cell-free RNA given its unique characteristics. The study employs a systematic approach. However, some methodological details are insufficiently described, which hinders full replicability. Specific concerns are as follows:
- Line 115-121: The threshold values used for identifying outliers are not provided. Supplemental Table S5 includes the (average?) values for each merged tissue type and the number of outliers, but it is unclear why these samples were excluded. Please provide details on the criteria used for filtering.
- Line 123: Similarly, the metric and threshold used to determine expression similarity when combining cell types are not specified. Including this information is crucial for reproducibility.
- Figure 6 & Supplemental Tables S2-S4: The process for calculating AUCs in Figure 6 is unclear. Were these values based on the validation set using a 7:3 test-validation split, or was a cross-validation or bootstrapping approach used? Additionally, the type of error bars in Figure 6 should be explicitly defined. To better represent variability, I recommend including a variability metric (e.g., standard deviation or confidence intervals) in Supplemental Tables S2-S4 alongside the currently listed (average?) AUC values.
- Line 110-111: I understand that retraining the model for tissue-of-origin analysis may require significant effort. However, given that many of the co-authors were involved in developing the cell-type deconvolution model, Deconformer, could they elaborate on why the hardware requirements exceed the capacity for training the tissue-of-origin model? Is it more computationally intensive than the cell-type model, or has available hardware capacity decreased?

2) Upon reviewing the reference, I found that Deconformer is a method developed by many of the same co-authors as this paper. However, this connection is not explicitly acknowledged in the text. For the sake of full transparency, I recommend using phrasing such as “our previously developed method” or “the method we developed” to clearly disclose the authors’ involvement.

3) To increase transparency, I also recommend making the code used for AUC calculations publicly available.

Validity of the findings

1) The manuscript encourages meaningful replication by clearly articulating its rationale and benefits to the field. The authors effectively compare the performance of various deconvolution approaches for cfRNA.

2) While the underlying data are sourced from other studies, the processing steps are described. However, explicit details about the quality filtering steps should be included to ensure reproducibility (see also comments above).

3) The conclusions are well-stated and linked to the original research question. However, I still have two remarks:
- The number of samples available for training and testing likely differs between datasets. Does this impact classification performance? Please also include the number of samples in the manuscript text or figures.
- Could you also add the output from Deconformer to Figure 2A? Although Deconformer could not be used for tissue-of-origin analysis in Figure 2B, including its performance in Figure 2A would provide an interesting comparison with other methods trained for cell-type deconvolution.

Additional comments

I commend the authors for conducting a systematic assessment of various deconvolution methods, particularly addressing challenges associated with plasma cell-free RNA. The approach to simulate cfRNA data based on cells and drop-out rates, while accounting for clinically relevant parameters, is elegant. However, there are areas that require further clarification or correction as stated above.

---

## Round 0.2 · Minor Revisions

Please address remaining comments made by the reviewer 2.

Reviewer 1 ·

Basic reporting

The authors responded all the comments from this reviewer with additional figure panels and modification of the text. There is no other comments.

Experimental design

Appropriately described.

Validity of the findings

Comparisons of the methods are valid and informative.

Additional comments

None.

Reviewer 2 ·

Basic reporting

My previous comments have been adequately addressed by the authors.

Experimental design

Most comments have been adequately addressed by the authors. Below are my remaining comments:

1) Tables S4-S6: Can you specify what sensitivity and specificity represent in these tables? Are these also the mean from the test sets of 100 models, and was the threshold fixed for all or optimized per model?
Typo in the column names of these tables: “desease” should be “disease”

2) Original comment reviewer:
Line 110-111: I understand that retraining the model for tissue-of-origin analysis may require significant effort. However, given that many of the co-authors were involved in developing the cell-type deconvolution model, Deconformer, could they elaborate on why the hardware requirements exceed the capacity for training the tissue-of-origin model? Is it more computationally intensive than the cell-type model, or has available hardware capacity decreased?
Response authors:
Thank you for your suggestion. By comparing the results of cell and tissue origin tracing from CSx, AutoGeneS, GEDIT, MuSiC, and SCDC, we observed that the proportion of whole blood is less than 20%, which is significantly lower than previous research findings and common sense. We speculate that the presence of many shared cell types across different tissues may lead to inaccuracies in the results, such as observing a higher proportion of spleen compared to whole blood. Therefore, when developing Deconformer, we did not train the tissue-of-origin model, not because the hardware requirements exceeded our capacity or because it was more computationally intensive.
Response reviewer:
I based my original comment on the following text in the manuscript: “Deconformer only provides a cell-type deconvolution model based on the TSP dataset. To perform tissue-of-origin analysis, the model would require retraining, which involves hardware requirements that exceed our capacity. Therefore, Deconformer was only used for cell-type deconvolution.”
However, the response here seems to contradict this. Can you please adjust the manuscript text accordingly?

Validity of the findings

Remaining comment:
I cannot see captions of the supplementary figures (not included in the provided PDF nor in document with tracked changes). Therefore, I could not evaluate whether the calculation method used for correlation was indeed properly added to the caption of Figure S2.

Additional comments

no comment

---

## Round 0.3 · accepted · Accept

Authors have addressed all of the reviewers' comments and the manuscript is ready for publication.